# Growing Brains, Nurturing Minds—Neuroscience as an Educational Tool to Support Students’ Development as Life-Long Learners

**DOI:** 10.3390/brainsci12121622

**Published:** 2022-11-26

**Authors:** Hagar Goldberg

**Affiliations:** Department of Psychology, University of British Columbia, Vancouver, BC V6T 1Z4, Canada; hagar.goldberg@ubc.ca

**Keywords:** educational neuroscience, neuroplasticity, human learning and development

## Abstract

Compared to other primates, humans are late bloomers, with exceptionally long childhood and adolescence. The extensive developmental period of humans is thought to facilitate the learning processes required for the growth and maturation of the complex human brain. During the first two and a half decades of life, the human brain is a construction site, and learning processes direct its shaping through *experience-dependent neuroplasticity*. Formal and informal learning, which generates long-term and accessible knowledge, is mediated by neuroplasticity to create adaptive structural and functional changes in brain networks. Since experience-dependent neuroplasticity is at full force during school years, it holds a tremendous educational opportunity. In order to fulfill this developmental and learning potential, educational practices should be human-brain-friendly and “ride” the neuroplasticity wave. Neuroscience can inform educators about the natural learning mechanisms of the brain to support student learning. This review takes a neuroscientific lens to explore central concepts in education (e.g., mindset, motivation, meaning-making, and attention) and suggests two methods of using neuroscience as an educational tool: teaching students about their brain (content level) and considering the neuro-mechanisms of learning in educational design (design level).

## 1. Educational Neuroscience (Teaching for the Brain and Teaching about the Brain)

*Educational neuroscience* is an interdisciplinary field exploring the effects of education on the human brain and promotes the translation of research findings to brain-based pedagogies and policies [1]. The brain is the target organ of education. Education is thought to influence brain development [2,3] and health, even as the brain ages [4,5]. Studying the dynamics between the brain and education can be instrumental in finding ways to better support learners across the lifespan.

Educational neuroscience research explores every possible relationship between the physiological, mental, and behavioral aspects of learning. Some studies have tried to identify the optimal physical conditions for neuroplasticity and learning. This stream of educational neuroscience research includes studies exploring the effects of sleep (or sleep deprivation), physical exercise, and environmental pollution on the brain and its cognitive performance [1]. While these studies focus on the effect of brain health on learning, other studies examine the effect of learning on brain health, assessing the long-term effects of learning/education on the human brain and exploring in what ways formal/informal education is associated with better aging of the human brain [2,3,4].

Some educational neuroscience studies take a developmental approach to study the relationship between cognitive and learning capacities across the lifespan. For example, multilevel measurements collected from adolescents (e.g., neuronal, hormonal, psychological, and behavioral) have advanced our understanding of how the massive neuronal changes that take place during adolescence promote cognitive development but also introduce immense neuronal and mental vulnerability (and the onset of most psychiatric disorders) [1,5,6,7]. Other studies in this line of research explore the factors supporting neuroplasticity in the mature brain—to support lifelong learning [8].

Educational neuroscience also explores the nature–nurture aspects of learning, for example, examining how learning environments interact with genetic conditions and what DNA variations predict differential learning abilities [9]. Environmental influences on learning include studies about the impacts of socio-economic status (SES) on the brain and cognitive developmental trajectory [10]. Furthermore, educational neuroscience seeks to understand the mechanisms that facilitate general learning abilities (such as executive control and social and emotional skills), discipline-specific learning abilities (such as literacy, numeracy, and science), the connections between these mechanisms, and the extent to which these learning skills are trainable [11].

As a developing, interdisciplinary research field, educational neuroscience faces challenges, limitations, and criticism, especially concerning the ability to generalize research findings in lab conditions to classroom learning, and its validity and transferability to larger scales, such as mass education systems. Other challenges stem from the fact that learning is one of the most basic yet complex brain functions that incorporates the entire brain and has a continuous effect. Furthermore, empirical studies in educational neuroscience are challenging and cumbersome due to the interdisciplinary nature of the field (education, psychology, and neuroscience); the need for repeated measures over time; and the young target population (school students), which imposes ethical restrictions on experimental designs. Finally, while still evolving as a research field, educational neuroscience is intriguing for many educational leaders who are enthusiastic about applying neuroscience in education practices. Unfortunately, the current gap between the high demand and limited supply may lead to misuse of neuroscience in pedagogy (e.g., neuromyths or the justification of educational methods based on limited to no evidence) [1].

While educational neuroscience is preliminary in forming evidence-based pedagogy, it can already offer valuable information and a much-needed bridge between educators and scientists in translating the research of learning into effective educational practices.

Neuroscience-informed educational design (teaching the way the brain learns) can promote learning motivation, high-level information processing, and knowledge retention. Moreover, neuroscience educational content (teaching about the brain) can inform students about their developing brains to promote scientific education and self-exploration.

### 1.1. Learning and Neuroplasticity

Human development is based on nature (genetics), nurture (physical and social environments), and their interactions (epigenetics) [12,13]. These factors play an essential role in learning processes and the reorganization of neuronal networks to create neuronal representations of new knowledge. Learning and training new knowledge or skills evoke specific and repeated activity patterns, and in the process of Hebbian neuroplasticity, neural pathways are reinforced by the strengthening of specific synapses, while less functional ones are eliminated [14,15,16].

Almost half a century ago, Vygotsky introduced the zone of proximal development (ZPD) [17] in education. According to the ZPD, learning and development depend on an optimal balance between support and challenge (see Figure 1: the zone of proximal development and neuroplasticity), which should be tuned and tailored for each learner based on their specific developmental stage. The ZPD model was revolutionary, as it emphasized the importance of the educational environment (nurture) in unlocking the internal potential (nature) of students, and it placed the learning *process* (as opposed to the learning *product*) as the central educational goal [17]. Some decades later, the biology of learning revealed a beautiful alignment with Vygotsky’s theory—with evidence showing that brain neuroplasticity is highly affected by environmental conditions and the balance between demands (challenge) and available resources (support) [18]. The impact of stressors on learning can be constructive or destructive depending on the intensity, duration, and accumulation of the stressors and the coping mechanisms and support that one has.

Neuroscience research suggests that experience-dependent neuroplasticity [19], which facilitates learning processes, benefits from several principles. The central one is that learning a skill or new knowledge requires the activation of relevant neuronal pathways. The research also points to the saliency, intensity, and repetition of the learned skill/knowledge as valuable strategies for enhancing neuroplastic changes [16,20,21]. Learners cannot be passive recipients of content but must be active participants in the learning process.

An enriched environment for enhanced neuroplasticity offers physiological integrity, cognitive challenge, and emotional safety. More specifically, an enriched environment includes adequate sleep and nutrition, sensory–motor and cognitive challenges, opportunities for exploration and novelty, and secured relationships that act like a safety net and enable learners to take on challenges [22,23]. Conversely, a lack of these conditions may slow down or decrease the level of neuroplasticity in the developing brain.

The social and cognitive safety net that enables learners to aim high while taking risks and to turn failure into resilience is rooted in safe relationships (with adults and peers) and in holding a growth mindset. A growth mindset is the belief that intelligence and learning potential are not fixed and can be developed [24]. Holding a growth mindset has been associated with academic success, emotional wellbeing, and motivation while reducing racial, gender, and social class achievement gaps [25,26,27,28,29,30]. While the impact of mindset interventions on academic performance is debatable regarding the general population [31], the literature is clear about the potential of growth mindset intervention in supporting the academic development of high-risk and economically disadvantaged students [26,27,31,32].

The notion of human potential as something dynamic resonates with the concept of the plastic brain. Moreover, teaching students about neuroplasticity and the dynamic potential of their brains has been shown to effectively reinforce a growth mindset [32].

#### 1.1.1. Using Neuroplasticity as Educational Content

Teaching students about experience-based neuroplasticity and the dynamic changes in neuronal networks during learning provides strong evidence of their natural and powerful learning capacity. Furthermore, teaching students about neuroplasticity with explicit connections to the growth mindset and development creates a motivating premise for learners—according to which their learning potential is dynamic and depends significantly on their attitudes and learning practices.

The neuroplasticity rules of “use it or lose it” and “use it to improve it” mean that, while teachers should support and guide them, learning occurs by and within the students. This physiology-based realization can help build students’ responsibility and ownership over their learning.

Harnessing neuroplasticity and a growth mindset to motivate students can be especially important with neurodivergent learners, whose cognitive development and learning styles deviate from the typical range. Twenty percent of the population is neurodivergent, including students on the autistic spectrum (ASD), students with learning disabilities (e.g., dyslexia), attention disorders (e.g., ADHD), neurological disorders (e.g., epilepsy), and mental illness (e.g., PTSD). While neurodiversity and variations in neuronal and cognitive expressions hold many advantages [33], neurodivergent students face extra challenges navigating neurotypical-oriented school systems. Learning about neuroplasticity can be a potent form of validation for neurodivergent students, as neurodiversity is a natural result of experience-dependent neuroplasticity [19]. In addition, by fostering a growth mindset and neuroplasticity awareness, neurodivergent students can be motivated to participate in evidence-based interventions. For example, teaching students with dyslexia about the specific structural and functional brain changes associated with the reading interventions that they apply [34,35,36] can motivate them to endure the hard work before noticing visible results.

#### 1.1.2. Using Neuroplasticity to Guide Learning Design

Organizing learning systems around conditions that promote neuroplasticity can enhance learners’ academic development and wellbeing. When a student accomplishes today what was not in their reach yesterday, it is the product of neuroplasticity through a growth mindset.

Educational environments that promote neuroplasticity include encouraging and modeling a healthy lifestyle (physical exercise, a balanced diet, sufficient sleep, and regulated stress), —for example, educating students about the counter-productiveness of sleep deprivation (e.g., “all-nighter” study marathons) on learning. In addition, learning systems should invest in intellectual stimulation (novelty and challenge) and the system’s social and emotional climate (human connections). Neuroplasticity and development are optimal in the stretch zone, where learners experience a motivating level of challenge and stimulation while feeling emotionally supported and socially safe. This ratio between support and challenge should be individualized (between learners and within learners over time).

Educating teachers about neuroplasticity can be powerful in understanding and supporting students that were affected by trauma. Childhood adversity hampers neuroplasticity duration and magnitude [37]; a surviving brain is not a learning brain. While neuroplasticity is compromised by early trauma, neuroplasticity is also the key to healing from trauma. Schools have a pivotal role in battling the damage of early trauma by creating enriched and safe learning environments that reinforce alternative neuronal pathways to reverse the effects of early adverse environments on child brain development [22,38,39,40].

### 1.2. Learning Motivation and Reward

Learning and adaptation are essential for surviving and thriving in dynamic environments. The brain evolved to make sense of information from our external and internal environments and to produce adaptive behaviors that promote survival. The brain is, therefore, a learning machine by nature, and learning does not require external initiation. However, learning is highly experience-dependent and can be directed and enhanced through education.

The brain reward system evolved to reinforce effortful behaviors that are essential for survival (e.g., foraging, reproduction, and caregiving). Such behaviors activate the dopaminergic system associated with reward and motivation [41]. The hormone/neurotransmitter dopamine is a central player in reward-motivated behavior and learning through the modulation of striatal and prefrontal functions [42]. The human brain reward system balances between (limbic) impulsive desire and (cortical) goal-directed wanting to guide flexible decision-making and adaptive motivational behaviors.

Psychologically, intrinsic motivation is driven by the need to experience a sense of competence, self-determination, and relatedness [43,44,45,46,47].

*Competence* refers to a perception of self-efficacy and confidence in one’s abilities to achieve a valuable outcome. *Self-determination* refers to the sense of autonomy and agency in the learning process. *Relatedness* refers to the drive to pursue goals that hold social value, which can be achieved by working collaboratively as part of a team or by creating something that resonates with others. Relatedness is a strong motivational driver, as it touches on a primary and primordial need to be part of a group and a higher spiritual and intellectual need for self-transcendence and impact.

Overall, these components are based on the human inclination to be valued and validated by the self and others. Biologically, they reflect basic survival needs that combine self-reliance (competence and ownership) and social reliance. Psychologically, these are all subjective perceptions that serve the need to maintain positive self-perception and self-integration. Finally, educationally, they reflect the natural human curiosity and tendency to learn and develop continuously.

The human brain reward system in the 21st century is an evolutionary mismatch. There is a discrepancy between the conditions that the reward system evolved to serve and those that it often faces in the 21st century. The reward system evolved over millions of years to motivate humans to work hard (invest time and energy) in maintaining their survival needs (e.g., nutrition, protection, reproduction, and the learning of new skills). However, this system is not designed for the abundance and immediacy of stimulation in the digital and instant reward era, which promotes the persistent release of dopamine that leads to an increased craving for reward (seeking behavior; wanting) and a decreased sense of pleasure and satisfaction (liking) [42,48].

Some of the most significant challenges of modern education systems relate to the massive changes in how people consume information and communicate in the digital era. Digital platforms have become dominant in information consumption and communication, which provide access to unlimited information and reinforce immediate rewards.

#### 1.2.1. Using Neuroscience (of Reward and Motivation) as Educational Content

The science of human motivation, including its evolutionary mismatch, can be utilized to shed some light on students’ struggles with learning motivation. It can further provide a framework for students to explore their motivational (approach or avoid) tendencies regarding learning and academic challenges. Moreover, learning the neuroscience underlying motivation and reward can raise students’ awareness and proactivity in managing and protecting their reward system. Since adolescence is the peak time for the initiation of substance use, and early onset imposes a higher risk of mental health and substance abuse disorders persisting into adulthood [49,50,51], neuroscience knowledge about the reward system and its vulnerability (especially during brain development) is essential educational knowledge that can help in the prevention and mitigation of teen addiction.

#### 1.2.2. Using Neuroscience to Guide Learning Design and Intrinsic Motivation

While students of the digital era are the most stimulation-flooded and attention-challenged in human history, learning is a process that takes time, selective attention, and perseverance. Therefore, learning designs that harness students’ intrinsic motivation for training and the development of stamina and grit (skills that might be hampered in the digital era) are precious for students’ health and success.

Motivational drivers include an adequate level of challenge that fits the student’s sense of competence and that creates optimal arousal levels, opportunities to expand social relatedness and impact, and balance between support and autonomy (see the ZPD, Figure 1).

Importantly, in classroom learning, educators are required to manage the attention, motivation, and reward system of not one but many students, which is a complex task. The typical classroom presents a broad spectrum of learners with diverse learning needs and stretch zones (Figure 1). While the facilitation of autonomy and the sense of competence varies between learners and requires personalized support, the social norms that promote learning are more ubiquitous and apply to most learners. While educators do not always have the resources to support students’ motivation individually, harnessing the social aspects of classroom learning is a manageable, effective strategy to elevate students’ motivation. Learning environments that demonstrate empathy, inclusiveness, and psychological safety have shown positive results in students’ behavior, self-esteem, motivation, and academic success [52,53,54,55,56]. Social motivation has been shown to enhance the encoding of new information (even if the content is not social) [57]. Learning-for-teaching and peer tutoring (one student teaching another student) effectively encode information into memory. Beyond memory improvement, peer tutoring has many further benefits to both the tutor and the learner in academic achievements [58,59], motivation, and ownership over the learning process and results in a deep conceptual understanding of the material [60].

The teacher’s demeanor is another controllable factor with a high potential to affect students’ motivation. For example, the literature points to teachers’ *immediacy* (creating physical and psychological closeness with students) as an effective way to enhance students’ engagement, learning motivation, and performance (including memory retention) [61,62,63,64,65]. Immediacy can be demonstrated through verbal and non-verbal gestures that communicate interest and personal connection (relating to personal stories, using animated voice and body language, creating eye contact, and using humor).

The research also indicates that, when students perceive the content as being personally relevant, they are more motivated to study [66]. Therefore, educators can actively make the learning content more relevant by using stories and real-life examples, making explicit connections and demonstrations of how the content may be relevant/applicable to the students, and giving students opportunities to reflect and share their connections to the learning material.

In summary, physiological and psychological approaches point to primary motivational drivers that direct engagement and investment in the learning process. Not surprisingly, these drivers that are anchored around social and intellectual needs align with the conditions supporting neuroplasticity discussed in the first part of this review.

### 1.3. Intrinsic and Extrinsic Processing in Learning and Meaning-Making

As the environment provides more information than the brain can handle, survival depends on saliency detection and attention management to direct perception and behavior. The brain constantly selects and attends to relevant input while suppressing irrelevant or distracting information [67]. Information that is valuable or urgent for survival and prosperity receives attention. Attention capacities (e.g., alerting, orienting, and controlling attention) are managed by several brain systems that interact and coordinate [68,69,70]. Top–down, cognitive-driven attention that fosters a goal-directed thinking process is associated with the dorsal attention network (consisting of the intraparietal sulcus and the frontal eye fields) [71]. This mechanism enables students to read a paragraph, listen to a lecture, think about the teacher’s question, or write an essay. A second attention system is bottom–up and stimulus-driven, and it orients attention to unexpected and behaviorally relevant stimuli. This ventral attention network consists of the right temporoparietal junction and the ventral frontal cortex [71]. This attention-grabbing mechanism enables the individual to respond quickly to urgent environmental demands, for example, moving away to prevent a struck-by-object accident. Flexible attention control depends on dynamic interactions and switching between the two systems and involves the central executive network (CEN) [68,70].

The insula and anterior cingulate cortex comprise the core structures of the saliency network [72], another major player in attention altering to emotionally salient stimuli through the interaction of the sensory and cognitive influences that control attention [72,73,74].

In addition to outward-focused attention, the human brain is also invested in inward-focused processing. Functional brain imaging studies of the human brain show a robust functional anticorrelation between two large-scale systems, one highly extrinsic and the other deeply intrinsic [75,76,77]. The central executive network (CEN) is an externally driven system and is paramount for attention control, working memory, flexible thinking, and goal-directed behavior. The core components of the CEN are the dorsolateral PFC and the lateral posterior parietal cortex (hence, the frontoparietal network) [72,78]. When the human brain is not occupied with external tasks, the default mode network (DMN) is activated. This internally driven cognitive network includes the posterior cingulate cortex (PCC) and the medial prefrontal cortex (MPFC) as core components. The DMN is thought to facilitate reminiscing, contemplating, autobiographical memory, self-reflecting, and social cognition [79]. Conversely, the DMN is immediately suppressed when the brain is engaged in externally driven tasks and stimulation.

Resting-state brain imaging studies revealed that the activity in the DMN during resting awake states indicates the quality of subsequent neural and behavioral responses to environmental stimuli [72,80]. Moreover, a high connectivity between “intrinsic” (DMN) and “extrinsic” (CEN) brain networks, and specifically emotional saliency, attention (extrinsic), and reflection (intrinsic) networks is associated with better cognitive performance, meaning-making, and broad perspective thinking [75,76,81]. These networks function antagonistically but are highly connected and balance each other. Furthermore, the anticorrelation between their function is associated with better task performance and positive mental health [79,82]. Recent studies also suggest that a causal hierarchical architecture orchestrates this anticorrelation between externally and internally driven brain activities. More specifically, that regions of the saliency network and the dorsal attention network impose inhibition on the DMN. Conversely, the DMN exhibits an excitatory influence on the saliency and attention system [79].

#### 1.3.1. The Neuroscience of Extrinsic and Intrinsic Processing as Educational Content

Teaching students about the dynamics of the default mode and executive control network can help them understand how their brain processes information, the importance of each process (e.g., extrinsic and intrinsic), and their integration for meaningful learning. This knowledge can be applied as students explore and experiment with ways to enhance their learning and memory by intentionally engaging both intrinsic and extrinsic processing and integrating the two.

#### 1.3.2. Using the Neuroscience of Extrinsic and Intrinsic Processing to Guide Learning Design

Traditionally, instructional education is based on learning objectives that are externally dictated and is focused on outward attention (stimulus-driven lectures and assignments). Mind-wandering has become the enemy of classroom teachers, as it indicates students’ lack of attention and poor learning.

Nevertheless, neuroscience research indicates that meaning-making and cognitive performance benefit from the interplay between extrinsic and intrinsic oriented attention and processing [56,75].

Learning instructions should consider the different attention mechanisms, evoking adequate arousal levels and leading to goal-directed thinking. Furthermore, students will benefit from an educational design that stimulates the natural interplay between “intrinsic” (DMN) and “extrinsic” (CEN) brain networks by incorporating external stimulation (e.g., presenting content), allocating time and space for intrinsic reflection (e.g., guided reflection and journaling), and integrating the two (e.g., guided class discussion and insights sharing) [83,84].

## 2. Discussion

### 2.1. Teaching Students about Their Developing Brains

As far as we know, humankind is the only species with access to the underlying mechanisms of its perception, learning, and inner workings. In addition, the human brain is endowed with a lengthy developmental period of approximately 25 years [85,86]. Therefore, schooling years are the prime time for neuroplasticity, and students can learn about their brains while they are highly malleable and can utilize this to amplify their learning and growth.

While students were traditionally required to choose whether to focus on the humanities or science fields, an integrative view is becoming increasingly common in academic institutes. Multidisciplinary studies have been shown to promote students’ positive learning and professional outcomes [87]. Teaching neuroscience from a dual perspective, both scientific/objective and humanistic/subjective, is a novel but natural bridge between the humanities and science fields.

Studying neuroscience with explicit connections to the lived experience of brain development and its behavioral manifestation can be academically and personally transformative for students.

Shedding a scientific light on students’ experiences as they unfold can support significant developmental processes during those years, such as improvements in executive functions, emotional regulation skills, meta-cognition, and social cognition [88].

Among the topics and burning issues of teens and young adults that neuroscience can offer insights into are selective and leaky attention [89], the reward system and addiction [90], the PFC–limbic developmental mismatch during adolescence [91], neurodiversity and inclusion, emotion regulation, and mental health [92].

Moreover, adding a personal layer to neuroscience studies fits the notion that personal relatedness and relevance are essential for learning motivation. Teaching neuroscience from a dual (scientific and personal) perspective and connecting neuroscience knowledge to a deeper understanding of the self and others can elevate engagement and nurture students’ passion for science and their ability to integrate and transfer scientific knowledge across contexts. In addition, similar to the effect of physical education, educational neuroscience can promote the awareness of brain health and encourage students to be intentional about their education and developmental trajectory.

### 2.2. Teaching the Way(s) the Human Brain Learns Best

Teaching students in brain-friendly ways means implementing principles that align with how the human brain encodes, consolidates, and retrieves information. Educational neuroscience points to the importance of a holistic and integrated view of cognitive, emotional, and social aspects to support learning and development [52,75,93,94]. Maintaining physical health, cognitive challenge, and emotional safety are essential factors in creating an enriched environment that supports neuroplasticity and learning.

Assessing the learning progress rather than the end product can encourage students to move away from rote memorization to more meaningful learning that carries on beyond the final exam.

Meaningful learning can be promoted by learning designs that encourage students to take experimental and explorative approaches, take risks, and make mistakes without detrimental consequences to their grades.

Furthermore, assessments throughout the learning process and not only at the end of it, using multiple sample points and low-risk tasks, can provide information on the student’s learning curve and allow for personalized and timely feedback that students can apply to improve their learning on the go.

These methods not only promote psychological safety but also align with the evidence-based practices of building long-term and accessible knowledge by spreading out the learning concept across time (spacing), practicing information retrieval from memory (recall), and integrating and transferring knowledge (the application of knowledge in different contexts) [95].

## 3. Conclusions

Brain knowledge is brainpower; teaching students about their developing brain can support their academic and personal development by deepening their understanding of science and humanities, their mental capacity, and their self-identity. Educational neuroscience is a promising field in teaching students about their brains and teaching them in brain-friendly ways to support them in becoming lifelong learners.

## Figures and Tables

**Figure 1 brainsci-12-01622-f001:**
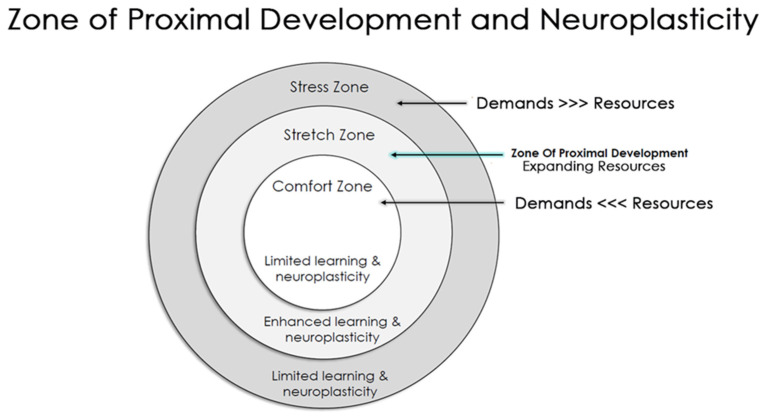
The zone of proximal development and neuroplasticity. An integrative approach between Vygotsky’s educational model and the neuroscience of learning. When learning and performance demands exceed the available support and resources, students are likely to be overwhelmed and resort to survival mode (stress zone). When learning and performance demands are significantly lower than the available support and resources, students are likely to be under stimulated and resort to static mode (comfort zone). When learning and performance demands match the available support and resources, students are likely to be appropriately challenged and work within their zone of proximal development, which promotes neuroplasticity and growth (stretch zone).

## Data Availability

The study did not report any data.

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
