# Peer review of "Growing Brains, Nurturing Minds—Neuroscience as an Educational Tool to Support Students’ Development as Life-Long Learners"

_brainsci, 2022, doi:10.3390/brainsci12121622_

Round 1
Reviewer 1 Report
The issue of the relationship between neuroscience and education is a fascinating one. The author attempts to summarise some of the relevant literature in this area, but the coverage is limited and rather superficial. There is no doubt that the reader would learn a great deal from this article. But what is lacking at the moment is a clear and coherent narrative or organising structure that makes the whole greater than the sum of the parts.
For me, the writing style was a significant barrier to comprehension. This might seem ironic, as much of the text is rather well written, but at a more general level, the writing lacks substance. In addition to some minor errors in writing style (such as changing tense), apparently inappropriate colloquialisms ("takes a lens on") and incomplete references, the manuscript reads rather like lecture notes or bullet points have been translated into prose. The frequent short paragraphs of a single sentence or two, and the lack of flow from one paragraph/section to another mean that this was not an easy read.
Author Response
I thank the reviewer for the honest feedback. A coherent narrative is critical in academic writing, especially when integrating different fields of research. Therefore, substantial text and content revision were done to form a more coherent narrative and a better flow.
Warmly,
Hagar Goldberg
Reviewer 2 Report
This is an extremely well written and well theorised paper. It draws upon recent theory and mostly critically dissects the contribution that neuroscience may make to learning and teaching. The literature, and the discussion are clear and based upon the research, so structurally and technically, the paper is clear and persuasive. However, there are three areas that the paper needs to address to make it better. The first is in the assertion that teachers need to ensure that students know that they are responsible for their own learning and progress, and that them being aware of neuroplasticity may help in this regard. Whilst that is undoubtedly the case for many students, there are a large number of students with particular special educational needs and different abilities who may well not be able to be so straightforwardly encouraged, nor motivated by established reward systems. These learners are not mentioned. Secondly, the paper does not address those learners who, by virtue of the fact of early childhood trauma, may not have the same levels of plasticity as other children. The impact of adverse circumstances thus needs to be acknowledged. Finally, the pedagogical implications of the area needs to be more critically examined. Some of the discussion on learning and teaching strategies seems rather superficial and a deeper discussion of fewer strategies but a deeper discussion of how they would work in practice, would be more useful.
Author Response
I would like to thank the reviewer for the constructive feedback which, I believe contributed significantly to the quality of this manuscript.
The reviewer highlighted three main points to be included in the paper:
The first two points were about the inclusion of neurodivergent learners in the paper. Neurodiversity and inclusion is an extremely important topic that deserves their own review. I included two new paragraphs in the first section of neuroplasticity, to address the important topic of neuro-minorities (and specific students with special needs and students who experienced childhood adversity.
I added in section 1.1.1 'Using neuroplasticity as an educational content –'
"Harnessing neuroplasticity and a growth mindset to motivate students can be especially important with neurodivergent learners, whose cognitive development and learning styles deviate from the typical range. Twenty percent of the population is neurodivergent, including students on the autistic spectrum (ASD), students with learning disabilities (e.g., dyslexia), attention disorders (e.g., ADHD), neurological disorders (e.g., epilepsy), and mental illness (e.g., PTSD). While neurodiversity and the variation in neuronal and cognitive expressions hold many advantages [32], neurodivergent students face extra challenges navigating the neurotypical-oriented school systems. Learning about neuroplasticity can be a potent form of validation for neurodivergent students, as neurodiversity is a natural result of experience-dependent neuroplasticity. In addition, by fostering a growth mindset and neuroplasticity awareness, neurodivergent students can be motivated to participate in evidence-based interventions. For example, teaching students with dyslexia about the specific structural and functional brain changes associated with the reading interventions they apply [32–34], can motivate them to endure the hard work before noticing visible results."
In section 1.1.2. Using neuroplasticity to guide learning design- I added the following:
"Educating teachers about neuroplasticity can be powerful in understanding and supporting students affected by trauma. Childhood adversity hampers neuroplasticity duration and magnitude [35]; a surviving brain is not a learning brain. While neuroplasticity is compromised by early trauma, neuroplasticity is also the key to healing from trauma. Schools have a pivotal role in battling the damage of early trauma by creating enriched and safe learning environments that reinforce alternative neuronal pathways to reverse the effects of early adverse environments on child brain development [36]."
In section 1.2.2 - Using the neuroscience of reward motivation to guide learning design and intrinsic motivation, I added the following:
"Importantly, in classroom learning educators require to manage the attention, motivation and reward system of not one, but many students which is a complex task. The typical classroom presents a broad spectrum of learners with divers learning needs and stretch zones (fig 1). While facilitation of autonomy and sense of competence varies between learners and requires personalized support, the social norms that promote learning are more ubiquitous and apply to most learners. While educators do not always have the resources to support students’ motivation individually, harnessing the social aspects of classroom learning is a managable, effective strategy to elevate students’ motivation."
The reviewer's third point was about the discussion of pedagogical implications.
The focus of this review is weaving together research perspectives from neuroscience, psychology, and education to shed new light on central concepts in education. Indeed the literature review and integration of the different perspectives receive more focus than the examination of pedagogical implications and therefore, some of the paper goals and titles were revised to reflect the content more accurately (as an important factor in the process of instruction and learning design).
In the abstract the premise was changed as follows:
"This review takes a neuroscientific lens to explore central concepts in education (e.g., mindset, motivation, meaning-making, and attention) and suggests two levels of using neuroscience as an educational tool: teaching students about their brain (content level), and considering the neuro-mechanisms of learning in educational design (design level)."
I thank again for the reviewer for the helpful review and hope I was able to address their concerns.
Warmly,
Hagar Goldberg
Reviewer 3 Report
Review of BRAINSCI-2020322: "Growing brains, Nurturing minds - Neuroscience as an Educational Tool to Support Students’ Development as Life-Long Learners"
Submitted to: Brain Sciences:
Summary:
This review article describes how neuroscience can have educational applications. First, educational neuroscience is defined, then the concept of neuroplasticity is applied to Vygotsky's zone of proximal development. Implications for learning are discussed. There is a discussion of motivation and reward, learning environments, and finally a concluding section on recommendations.
Overall evaluation:
I enjoyed reading this article and feel that it makes a useful primer to educational neuroscience and how it can be applied in practical ways to enhance student learning. It contains interesting insights and from what I can tell should be highly suitable for the relevant special issue. Overall, I did not find any major issues that would impede eventual publication. Below I mention a few areas where the author may consider making some improvements.
Additional comments:
1. The review could benefit from more thoroughly acknowledging the controversy over growth mindset, with some studies showing very minimal benefit of growth mindset interventions (e.g., Sisk, V. F., Burgoyne, A. P., Sun, J., Butler, J. L., & Macnamara, B. N. (2018). To what extent and under which circumstances are growth mind-sets important to academic achievement? Two meta-analyses. Psychological Science, 29(4), 549-571.).
2. The mention of evidence-based practices of enhancing knowledge includes spacing, but misses retrieval practice which has an equally strong evidence base. Out of all such learning techniques, spacing and retrieval practice both arguably warrant mention (e.g., see Carpenter, S.K., Pan, S.C. & Butler, A.C. (2022) The science of effective learning with spacing and retrieval practice. Nat Rev Psychol 1, 496–511).
3. The article already does touch on limits of the applicability of educational neuroscience given the nascent state of some aspects of the field, but those limitations could be expounded upon further.
Author Response
I would like to thank the reviewer for the thoughtful and constructive feedback. I addressed the reviewer's comments to the best of my ability.
1. The reviewer made an essential point about the controversy regarding the relationship between mindset and academic achievement (and especially the level of impact/potential of mindset interventions on academic performance. Therefore, a section in the paragraph about mindset and neuroplasticity was added as follows :
"While the impact of mindset interventions on academic performance is debatable regarding the general population [31], the literature is clear about the potential of growth mindset intervention in supporting the academic development of high-risk and economically disadvantaged students [26,31,32]."
2. The reviewer commented that the review does not mention retrieval practice, an evidence-based effective method for learning and producing long-term, accessible knowledge. Indeed research on human memory processes such as encoding, consolidation, and retrieval is critical to advance educational implications. As such, this line of studies was among the first connections bridging brain science and education to create the educational-neuroscience field.
Therefore, since the concept of memory and retrieval was vastly studied and reviewed by others, this review was focused on other, less discussed concepts (like the connection between neuroplasticity to the zone of proximal development, motivation and the interplay between intrinsic and extrinsic meaning-making). Still, retrieval is such a fundamental concept in learning and memory that it should be mentioned, even concisely, in this review. Therefore, a section in the paragraph about mindset and neuroplasticity was added as follows (including a reference to the implications of these concepts in educational design):
"These methods not only promote psychological safety but align with evidence-based practices of building long-term and accessible knowledge through spreading out the learning concept across time (spacing), practice information retrieval from memory (recall), knowledge integration and knowledge transfer (the application of knowledge in different contexts [86]."
3. The reviewer pointed out that although applicability limitations (of educational neuroscience) are mentioned, they could be expounded upon further.
The focus of this review is weaving together research perspectives from neuroscience, psychology, and education to shed new light on central concepts in education. Indeed, the integration of different research perspectives is discussed in more detail than the limitations of the application. The premise of the review, as described in the abstract, was therefore changed as follows.
"This review takes a neuroscientific lens to explore central concepts in education (e.g., mindset, motivation, meaning-making, and attention) and suggests two levels of using neuroscience as an educational tool: teaching students about their brain (content level), and considering the neuro-mechanisms of learning in educational design (design level)."
I thank again for the reviewer for the helpful review and hope I was able to address their concerns.
Warmly,
Hagar Goldberg
Round 2
Reviewer 1 Report
The author has responded constructively and comprehensively to the feedback from reviewers. The article is now much more coherent. In my opinion, it warrants acceptance in its current form.
Reviewer 2 Report
Thankyou for your corrections and amendments to the previous work.